# A cosmogenic $^{10}$Be anomaly during the late Miocene as independent time marker for marine archives

Dominik Koll [1,2,3] ✉, Johannes Lachner [1], Sabrina Beutner[4], Sebastian Fichter [1], Silke Merchel[1,5], Georg Rugel [1], Zuzana Slavkovská[2], Carlos Vivo-Vilches [1,5], Stella Winkler [1] & Anton Wallner[1,3]

Cosmogenic nuclide dating relies on the constancy of production and incorporation of radionuclides in geological archives. Anomalous deviations from constancy during the Holocene or Pleistocene are frequently used as global benchmarks to harmonize different data sets. A similar dating anchor on the million year timescale was so far not presented. In this work, we report on a prolonged cosmogenic $^{10}$Be anomaly during the late Miocene recorded in several Central and Northern Pacific deep-ocean ferromanganese crusts in the time period 9–11.5 Myr ago peaking at 10.1 Myr. Potential origins of this anomaly are discussed in the light of geological, climatic, solar and astrophysical events. This anomaly has the potential to be an independent time marker for marine archives.

The floor of the major oceans on Earth exhibits one of the most pristine geological archives recording millions of years of environmental conditions and changes, ferromanganese crusts. Dating of these marine archives can be accomplished through fossils by biostratigraphy[1], isotopic or elemental composition changes[2–4], or analysing the imprinted changes of Earth's magnetic field by magnetostratigraphy[5]. Another commonly employed technique is cosmogenic nuclide dating.

The radionuclide $^{10}$Be is continuously produced in the upper atmosphere by cosmic ray spallation mainly on nitrogen and oxygen[6–8]. The residence time of $^{10}$Be in the atmosphere is on the order of 1–2 yr until it attaches to aerosols and precipitates[9,10]. In the ocean, the atmospheric $^{10}$Be mixes with lithospheric stable $^9$Be, which is mainly transported into the ocean by river runoffs[11] and fluvial dust[12] after erosion of terrestrial minerals[13]. In-situ production of $^{10}$Be in marine archives as well as in terrestrial minerals is negligible compared to atmospheric $^{10}$Be due to atmospheric shielding and/or aquatic overburden. Therefore, atmospheric $^{10}$Be is dominating the $^{10}$Be inventory of the ocean. Marine archives, such as deep-sea sediments

and ferromanganese encrustations, incorporate $^{9,10}$Be into their matrix[14,15]. The incorporation and subsequent decay of cosmogenic $^{10}$Be ($t_{1/2}$ = 1.39 Myr[16,17]) is used to date marine accumulations on timescales from 100 kyr to 15 Myr ago by accelerator mass spectrometry (AMS), see ref. 18 for the most recent comprehensive summary on AMS. Cosmogenic $^{10}$Be dating is regularly applied to various geological archives ranging from Arctic and Antarctic ice cores[19] to deep-sea sediments[14] and ferromanganese crusts[20–23] as a consequence of its long-term stability and accuracy as well as the capability to detect small deviations from constancy through high-precision AMS measurements. Deviations from the pure decay curve of $^{10}$Be in marine archives are induced by changes in sedimentation or growth rate of the archive due to changing environmental conditions such as pH, trace elemental composition of seawater, scavenging effects, water currents, etc. Similarly, a deviation could occur, if the $^{10}$Be production rate changes due to variations in the cosmic ray flux[24]. Well-known deviations from constancy such as the Laschamp-event (multicentennial)[25,26] or Miyake-events (1 year)[27,28] are universal independent time markers with invaluable importance in geological and

[1]Accelerator Mass Spectrometry and Isotope Research, Helmholtz-Zentrum Dresden-Rossendorf, Dresden, Germany. [2]Department of Nuclear Physics and Accelerator Applications, The Australian National University, Canberra, Australia. [3]Institute of Nuclear and Particle Physics, TUD Dresden University of Technology, Dresden, Germany. [4]Institute of Resource Ecology, Helmholtz-Zentrum Dresden-Rossendorf, Dresden, Germany. [5]University of Vienna—Faculty of Physics, Vienna, Austria. ✉e-mail: d.koll@hzdr.de; dominik.koll@anu.edu.au

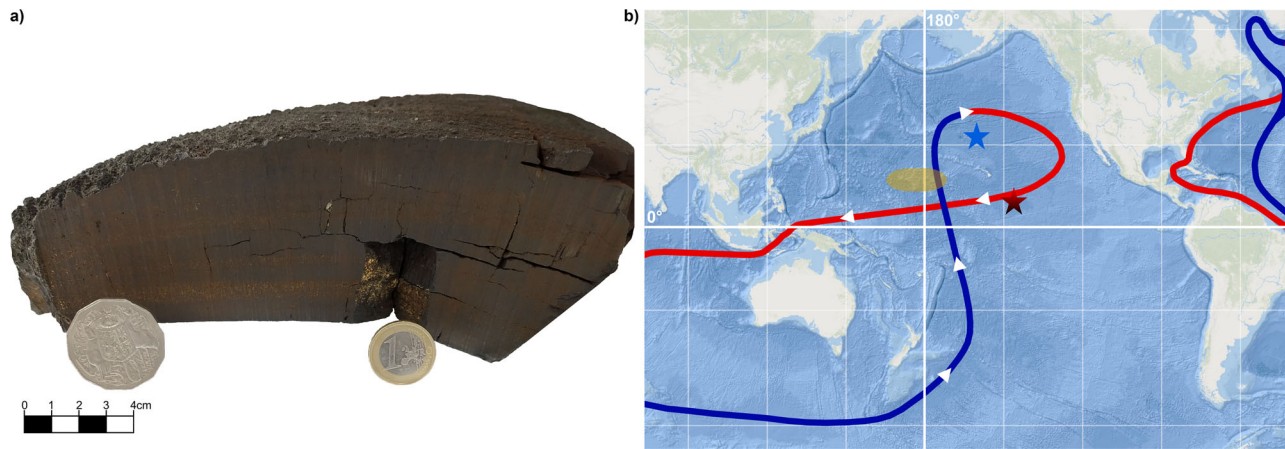

**Fig. 1 | Ferromanganese crusts from the Pacific Ocean. a** Photo of the ferromanganese crust VA13/2-237KD. A 1 euro coin and a 50 Australian cents coin are used as size references. **b** Locations of the ferromanganese crusts VA13/2-237KD (red star)[20,21,32], SO142-4DR (blue star)[35] and Crust-3[33] (yellow shaded area, exact location unknown due to resource protection). The major bottom (blue line) and surface (red line) ocean currents of the thermohaline circulation are indicated. The oceanic map was generated using Esri ArcGIS, Credit: Esri, GEBCO, Garmin, NaturalVue.

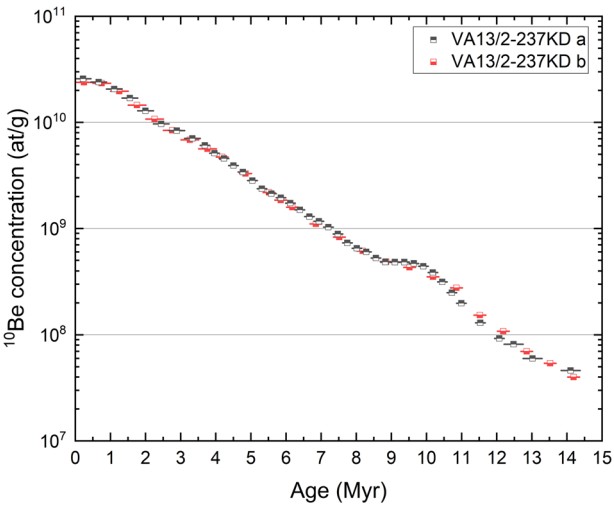

**Fig. 2 | $^{10}$Be concentration vs. age measured in crust VA13/2-237KD.** The profiles (**a**) and (**b**) were dated based on the determined growth rates of the ferromanganese crust. The anomalous increase of the $^{10}$Be concentration could be dated to the same age interval of 9–11.5 Myr ago.

archaeological dating, see e.g. refs. 29,30. A corresponding cosmogenic time marker for deep-ocean samples on the million year timescale would allow accurate dating of marine archives from the Pliocene to the Miocene up to more than 15 Myr ago using the cosmogenic radionuclide $^{10}$Be. Deep-ocean samples showed so far fairly constant $^{10}$Be levels through their $^{10}$Be concentration as well as $^{10}$Be/$^{9}$Be ratio profiles over several Myr[14,15,20–23,31].

Here, we report on the discovery of an anomaly in the $^{10}$Be concentration profiles of several deep-ocean ferromanganese crusts from the Central and Northern Pacific during the late Miocene.

## Results and Discussion

The determined $^{10}$Be concentrations in drill-holes (a) and (b) of the ferromanganese crust VA13/2-237KD decrease exponentially with depth as expected from radioactive decay (Supplementary Fig. 1). Two distinct growth periods with different growth rates can be distinguished between both drill-holes (a) and (b) for the depth intervals 2–8 mm and 8–28 mm, and 2–8 mm and 8–23 mm, respectively, as already discussed in ref. 32. The difference in growth between (a) and

(b) reflects the geometry of the crust; (b) was sampled at the slanting side, whereas (a) was taken from the flat centre (Fig. 1)[32]. The surface concentrations show a well-known flattening[20–23,31–33], which could be explained by an open-system exchange of Be with seawater and a resulting reduction of Be in the surface of the crust before equilibrium. Ages can be reliably calculated by fitting the concentration profile and, hereby, averaging fluctuations in growth of these natural samples. The ages are extended towards the surface by extrapolating the growth rates. No erosion of the surface is discernible from high-resolution visual 3D and micro-CT scans. The surface $^{10}$Be concentrations agree with surface concentrations of other ferromanganese crusts[15]. The dated concentration profiles for both drill-holes overlap nicely and show exemplarily the robustness of the procedure (Fig. 2). This piece of the ferromanganese crust could be dated to more than 18 Myr for a thickness of 50 mm by extrapolating a constant growth rate beyond 12 Myr ago despite a flattening in the profile. The gradual flattening of the $^{10}$Be profile towards deeper layers might be induced through an observed structural change of the crust matrix[34] in addition to a difference in growth rate. The full original crust had a thickness of up to 400 mm[34] with the start of growth happening well before 20 Myr ago.

### Presence of an anomaly

The concentration profiles deviate significantly from the expected exponential decay for the intervals 29–36 mm and 23–33 mm for drill-hole (a) and (b), respectively. These different depth intervals, however, translate to the same age interval of 9–11.5 Myr ago in this ferromanganese crust. The age interval was conservatively estimated to include the timing in both profiles. This anomalous increase in $^{10}$Be concentration appears to be synchronous. Sample intrinsic effects such as $^{10}$Be-rich inclusions in the form of micrometeorites as well as pore-water effects through cracks can be excluded due to the about 20 cm lateral distance of the two drill-holes as well as the different depth but same timing of this anomaly. Normalization to the matrix of the crust, the Fe concentration of the crust or the Be concentration of the crust only affects the amplitude of the anomaly but not its presence (Supplementary Fig. 2). To exclude a local effect at the crusts location, a $^{10}$Be concentration profile of the crust SO142-4DR from the North Pacific at a distance of about 2900 km from VA13/2-237KD was determined. This incomplete profile shows an anomaly at 16–18 mm depth (Fig. 3). The surface concentration in SO142-4DR is slightly lower but in fair agreement with the surface concentration of VA13/2-237KD and other Pacific ferromanganese crusts[15]. The $^{10}$Be concentration at the depth of the anomaly in SO142-4DR is the same as in VA13/2-237KD.

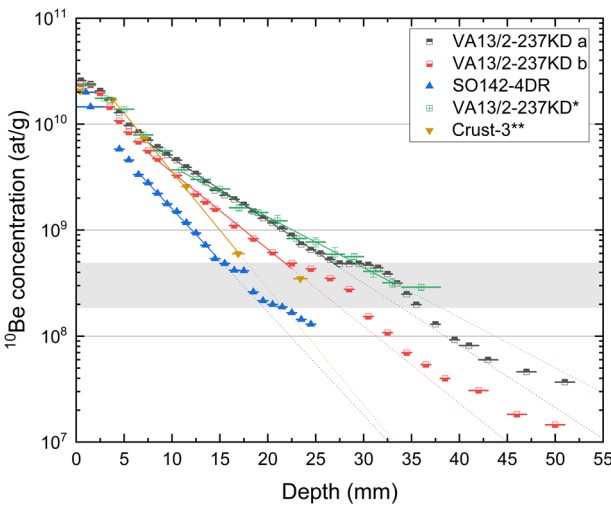

**Fig. 3 | $^{10}$Be concentration vs. depth measured in crusts VA13/2-237KD[20,21,32], SO142-4DR[35] and Crust-3[33].** The grey band corresponds to a similar age interval, where the $^{10}$Be anomaly appears in all data sets at different depths in the ferromanganese crusts, consistent with different growth rates of the crusts. * The $^{10}$Be concentrations were taken from the literature after re-analysis[20,21]. ** The $^{10}$Be concentrations were calculated from the $^{10}$Be/$^{9}$Be ratios taken from the literature[33].

Assuming similar surface concentrations, the same concentration where the anomaly occurs would translate to the same age of the crusts at the time of the anomaly. An exact age determination, however, would require a complete $^{10}$Be concentration profile. An independent dating of SO142-4DR was previously achieved by a $^{53}$Mn/$^{55}$Mn profile[35]. The determined growth rate of 1.52 mm/Myr would date the depth interval of the anomaly to 10.5–11.8 Myr ago, in perfect agreement with the age interval in VA13/2-237KD. It has to be concluded that the $^{10}$Be anomaly is imprinted into two different ferromanganese crusts from the Central and Northern Pacific Ocean around 10 Myr ago.

Since $^{10}$Be dating is regularly used for deep-ocean archives, an extensive literature review and re-analysis might affirm or confute this anomaly. The ferromanganese crust VA13/2-237KD was previously studied for the influx of interstellar supernova-produced $^{60}$Fe[22,31,32,36] and $r$-process $^{244}$Pu[31,32,37]. Initially, the dating was achieved by $^{10}$Be in 1984 right after retrieval of the crust[20]. The re-evaluated data set[21] is in agreement with the recently acquired ones (Fig. 3) with more fluctuations in the data. The last data point at 34–38 mm is enhanced with respect to the predicted exponential decay. A slightly deeper depth interval for the anomaly compared to (a) and (b) is in agreement with the structure of the crust and the sampling. The drill-hole taken by Segl et al.[20] was more centrally located than (a) and (b) and should therefore be more similar to (a) from the flat portion of the crust than to (b) from the slanting side, which is the case. The last data point can be seen as a hint for the anomaly. In conclusion, the enhancement of the last data point agrees with the more precise and extensive data on the $^{10}$Be anomaly of this work.

The investigation of another ferromanganese crust, Crust-3, for interstellar radionuclides[33] in 2021 also required cosmogenic $^{10}$Be dating. The $^{10}$Be/$^{9}$Be ratio of six drill-hole samples was used to interpolate the age of the 24 mm drill-hole profile of the crust. Recently, this profile was re-evaluated[31] and a $^{10}$Be concentration profile was generated (Fig. 3). This $^{10}$Be concentration profile also shows the well-known flattening towards the recent surface of the crust and remarkably an anomalously high $^{10}$Be concentration in a deeper layer (Fig. 3). The anomalous last data point at 22.7–24.0 mm is datable to 10.8–11.4 Myr ago using a re-evaluated average growth rate of 2.1 mm/Myr, and 9.5–10.0 Myr ago using the growth rate of 2.39 mm/Myr of Wallner et al.[33] which did not exclude the anomalous data-point. Hence, a $^{10}$Be

anomaly is also present in this data set making it the third independent Pacific crust sample with a strong indication for a $^{10}$Be anomaly around 9–11.5 Myr ago.

Tantalizing evidence for an anomalously high $^{10}$Be concentration was reported in 1985 in one of the first investigations of $^{10}$Be concentrations in marine sediments[38]. The North Pacific sediment cores DSDP 576 (32° N, 164° W) and DSDP 578 (33° N, 151° W) were analysed at the McMaster University for $^{10}$Be accumulation. After re-analysis, the anomalous data point in core 576 can be dated to 9.7 Myr ago when considering accumulation rates of 16.7 m/Myr in the interval 0–15 m and 1.7 m/Myr in the interval 15–30 m. However, the anomalous data point stems from a different drill-core compared to the preceding data points, which renders the comparability questionable despite efforts in harmonizing the timing of both drill-cores. Nevertheless, the clear enhancement at the right time period can be seen as a hint to a corresponding anomaly in deep-sea sediments. A further substantiation of a $^{10}$Be anomaly in drill-core 578 is not discernible due to the lack of time-resolved data.

The constancy of $^{10}$Be deposition was also investigated in two ferromanganese crusts[39]. The equatorial Atlantic crust K-9-21 (7° N, 21° W) and the North Pacific crust SCHW-1D (30° N, 140° W) showed good constancy in $^{10}$Be deposition over the last 7–9 Myr. An increase of $^{10}$Be concentrations beyond 7–9 Myr ago was discussed. The gradual increase of $^{10}$Be in SCHW-1D around 6 Myr ago, though, can similarly be described by a changing growth rate of the ferromanganese crust. The continual change in slope in the $^{10}$Be concentration graph of SCHW-1D in contrast to a peak-like structure is in agreement with this assessment. The coarse data obtained from K-9-21 do not allow a detailed assessment of a $^{10}$Be anomaly.

Lastly, a large compilation of ferromanganese crust and nodule $^{10}$Be data sets[40] was checked for any indication or an argument against a Pacific-wide or even global $^{10}$Be anomaly in the time interval 9–11.5 Myr ago. Only the already investigated data set of ref. 20, which was reproduced in ref. 40, and the profile of the ferromanganese crust 72 DK 9 need to be considered, because none of the other thirteen data sets covers the relevant time period. In particular, the promising data of the ferromanganese nodule DK 143 turned out to deviate from continuous growth in the time-period of the anomaly but only due to the transition from the authigenic nodule matrix to the inner centre rock substrate after re-analysis and re-fitting. The $^{10}$Be concentration profile of the ferromanganese crust 72 DK 9 shows strong statistical fluctuations around the fitted profile below a depth of about 25 mm due to the low $^{10}$Be concentrations and probably the lower AMS sensitivity at that time. The anomaly would be present at a depth interval of about 24–32 mm, where an enhancement of $^{10}$Be cannot be deduced anymore due to strong fluctuations. The $^{10}$Be anomaly is, therefore, neither clearly proven nor disproven in the discussed literature, but there are several indications supporting the presence of a $^{10}$Be anomaly.

## Amplitude and timing of the anomaly

Strong indications for a $^{10}$Be anomaly in the time period 9–11.5 Myr ago were found in three independent Pacific ferromanganese crusts with a total of five individual data sets. Any investigation of its origin requires the quantification of the surplus of $^{10}$Be as well as its timing. The high-resolution $^{10}$Be concentration profile from drill-hole (a) (Fig. 2) of the ferromanganese crust VA13/2-237KD can be decay-corrected based on the well-known $^{10}$Be half-life of 1.39 Myr[16,17] and the established growth rate of the crust[32]. The profile should then become linear if only the decay of incorporated $^{10}$Be leads to changes in its concentration. The decay-corrected $^{10}$Be concentrations are then normalized to the extrapolated mean equilibrium surface concentration of $3.6 \times 10^{10}$ at/g. A correlated fluctuation around the equilibrium surface concentration is visible during the age interval 0–9 Myr ago (Fig. 4). These sinusoidal periodic low-amplitude fluctuations could originate through density

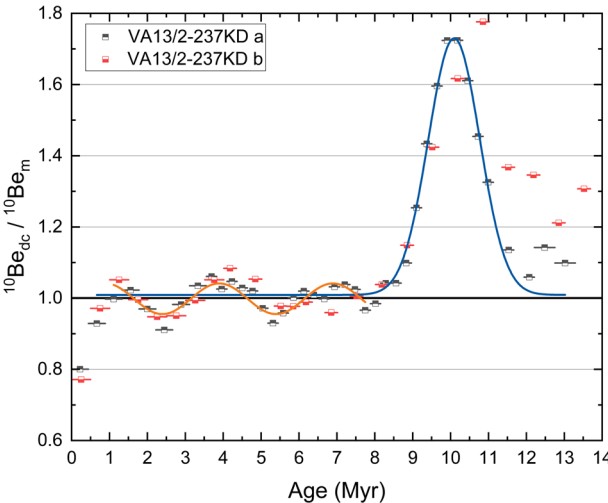

**Fig. 4 | The dated $^{10}$Be concentration profiles of VA13/2-237KD after decay correction (dc).** The extrapolated mean equilibrium surface concentration of $3.6 \times 10^{10}$ at/g was used to normalize the decay-corrected profile. The complete drill-hole profile (a) is used to identify any underlying structures, whereas the incomplete drill-hole profile (b) from the slanting side serves as an internal check. A sinusoidal fluctuation (orange line) around the surface concentration is visible during the age interval 0–9 Myr ago. A strong enhancement of $^{10}$Be concentration, a $^{10}$Be anomaly, is apparent in the age interval 9–11.5 Myr ago. The peak of the anomaly is fitted by a Gaussian (blue line) to 10.1 Myr ago with a full width at half maximum (FWHM) of 1.4 Myr.

and elemental composition changes of the ferromanganese crust due to its growth or cyclic ocean perturbations[41]. The reduction of the normalized $^{10}$Be concentration at the surface of the crust reflects the flattening of the $^{10}$Be concentration profile and the non-equilibrium condition at the crust-water boundary. The anomaly is visible as a strong increase of the $^{10}$Be concentration over the baseline at around 9 Myr ago and a return to the baseline beyond 11.5 Myr ago (Fig. 4). The decay-corrected and normalized $^{10}$Be concentration profile from drill-hole (b) shows a similar structure with a shift to older ages by less than 0.4 Myr, which can be explained by a slight deviation in the dating as well as a stronger variability due to the profile location at the slanting side of the crust. The peak of the anomaly in drill-hole (a) is fitted by a Gaussian to 10.1 Myr ago with a full width at half maximum (FWHM) of 1.4 Myr. The amplitude of the Gaussian represents a 73% increase (factor 1.73) compared to the baseline, whereas the sinusoidal fluctuations are 4% around the baseline. The $^{10}$Be inventory is higher by 25% in the time period 9–11.5 Myr ago compared to the baseline at younger ages.

The origin of this anomaly is yet unknown and in the following conceivable scenarios are discussed that could lead to such a $^{10}$Be anomaly.

## Geomagnetic field and solar activity variations

Production rates of radionuclides from reactions of galactic cosmic rays in the atmosphere are subject to changes in the geomagnetic field or in solar activity. A low geomagnetic field leads to a predicted doubling of the production rates compared to an average geomagnetic field strength[42]. Even during the most dramatic geomagnetic events, the recorded increase in $^{10}$Be is slightly less than a factor of 2 over periods of less than 10 kyr, e.g. for the Matuyama-Brunhes reversal[43] or the Laschamps excursion[26]. Paleomagnetic records[44–46] indicate a slightly lower geomagnetic field but show no extraordinary clustering of geomagnetic events during the period of the $^{10}$Be anomaly in the late Miocene. Repeated geomagnetic dipole-lows linked to reversals or geomagnetic excursions would need to be present in a unique phase of geomagnetic instability over many 100 kyr in order to produce a

prolonged signal. A prolonged series of dipole-lows would also be imprinted into the palaeomagnetic profiles of other oceanic archives.

Grand minima in solar activity, such as the Maunder or Dalton minimum, occur when multiple solar cycles have superimposed minima[47]. They typically last for decades or centuries and can be identified in records of $^{14}$C and $^{10}$Be in the Holocene[48]. Again, the cosmic ray intensity hitting Earth's atmosphere is increased too little and for too short periods to explain the observed overproduction recorded in the crust. Stronger excursions of natural radionuclide production in $^{10}$Be, $^{14}$C and $^{36}$Cl are only documented on shorter time scales during extreme solar events with high fluxes of solar protons with energies above 100 MeV. Such events have been documented to produce a 3-fold increase over the period of one year in the most prominent case of the AD 774/5 event[27,28,49], and slightly stronger for the event 9125 BP near a solar minimum[50].

In summary, the established mechanisms of radionuclide production within the atmosphere by cosmic rays considering geomagnetic field shifts, grand minima in solar activity or solar proton events can only explain an overproduction of $^{10}$Be on time-scales significantly below 100 kyr.

## Climatic and oceanic alterations

In previous work, major changes in the global abyssal circulation with reference to the onset of the modern global circulation were considered to produce $^{10}$Be anomalies without further explanation[38,39]. Indeed, during the Miocene, the oceans were significantly different compared to today. Mid-Miocene sea-level oscillations of 40–60 m[51] occurred due to the partial but rapid melting of Antarctic ice on timescales of <100 kyr[52]. A global sea-level rise induced by the melting of Antarctic ice containing long-term accumulated $^{10}$Be could increase the $^{10}$Be budget of the oceans. Assuming a mean ocean depth of 3700 m[53] with an average $^{10}$Be concentration of $1.4 \times 10^3$ at/g[54] and a 50 m sea-level rise induced by Antarctic melt-water with a $^{10}$Be concentration of up to $5.5 \times 10^4$ at/g[55], the total $^{10}$Be inventory of Earth's oceans would instantly increase by about 50%. Even higher enhancements can be achieved locally due to transport and scavenging effects. However, the $^{10}$Be budget would only gradually increase over the timescale of the melting event which is believed to be on the order of 100 kyr or less and reach a maximum far below the projected 50% due to dilution. Furthermore, the climate is in a cooling phase towards the end of the Miocene, see ref. 56 for a comprehensive summary about the Miocene climate and biota. After the Middle Miocene Climatic Optimum (MMCO), the global temperature is decreasing during the Middle Miocene Climatic Transition (MMCT) around 14 Myr ago[57–59]. Several intense glaciation events (Mi-events) were discovered in $\delta^{18}$O deep-sea sediment records[60]. The Mi-6 event occurs at around 10.4 Myr ago and the $\delta^{18}$O record reaches modern values indicating more stable but cooler climatic conditions. A sea-level fall of 50 m in the time-period 11.4–13.6 Myr ago[61] is further supporting this assessment. A strong enhancement of $^{10}$Be in the ocean because of additional input through the melting of Antarctic ice seems therefore implausible.

An enhancement of $^{10}$Be in the Pacific Ocean can be realized by the transport of $^{10}$Be-rich seawater through changing ocean currents or dissolution of $^{10}$Be-bearing deep-sea sediments. In the time period 9–12 Myr ago, the carbonate crash, a strong reduction of the calcium carbonate concentration in deep-sea sediments, occurred[62]. This event might have been triggered by a reduced surface-water productivity, a dissolution by corrosive deep-water, a dilution with carbonate-poor sediment or a biogenic bloom, see ref. 63 for a recent review and refs. 64,65 for in-depth discussions of potential causes. The potential causes are linked through a predicted change in global circulation patterns. Recently, the onset and ramp-up of the modern Antarctic Circular Current (ACC) was dated to occur during the time period 10–12 Myr ago[66]. The onset and ramp-up of the modern ACC coinciding with the timing of the $^{10}$Be anomaly and the provenance of the

ferromanganese crusts along the Pacific thermohaline circulation loop (Fig. 1) is striking. Seawater [10]Be concentrations vary on a global scale due to latitudinal production rate differences, atmospheric mixing and removal processes as well as residence and mixing times in the ocean[15,67,68]. The [10]Be concentration in seawater varies by more than a factor of 2 at recent times[15]. Therefore, an enhancement of the [10]Be concentration due to major changes in the ocean circulation pattern cannot be excluded. Stable element analysis of the ferromanganese crust VA13/2-237KD shows a strong decrease of the lithospheric [9]Be concentration towards recent times consistent with a major change of the trace element content of seawater (Supplementary Fig. 3).

Corrosive deep water was brought forward to potentially contribute to the reduction in calcium carbonate of deep-sea sediments during the carbonate crash. Assuming a comparable mechanism that leaches Be out of deep-sea sediments, the [10]Be concentration in seawater would increase. However, the [9]Be concentration should similarly increase, which is contrary to the decreasing [9]Be concentration in the ferromanganese crust (Supplementary Fig. 3). Most of the leached Be would eventually end up in the deep-sea sediment again due to the insignificant uptake of Be in ferromanganese crusts compared to the overall Be budget of the ocean.

In summary, the melting of Antarctic ice seems to be an unlikely scenario on its own, the contemporaneous onset and ramp-up of the ACC leading to a major re-organization of oceanic circulation is a viable scenario and corrosive deep water leaching requires a local strong decrease of [9]Be concentration to balance the increase of both Be isotopes by leaching.

Very recently, a 2.4 Myr eccentricity grand cycle was proposed to be a driver for deep-water circulation and erosive bottom current activity[41]. The proposal of cyclic deep-ocean disturbances by orbital dynamics is intriguing considering the sinusoidal [10]Be concentration profile oscillations and the larger [10]Be anomaly. Further investigations are required to define the effects on marine archives more precisely.

## Astrophysical events

In contrast to a constant atmospheric [10]Be production rate and geological reasons for a [10]Be anomaly in ferromanganese crusts, the production rate could be enhanced by either a changing atmosphere or a higher galactic cosmic ray (GCR) flux. A changing atmosphere, however, is a highly unlikely scenario, because only if a large fraction of the atmosphere changes from nitrogen to carbon, the production of [10]Be would be enhanced[69]. Even though volcanoes have the potential to increase the $CO_2$ levels on Earth[70], the required major change of Earth's atmospheric composition is unthinkable.

The remaining scenarios include a higher GCR flux due to astrophysical events. Recently, the possibility of a compressed heliosphere beyond Earth's orbit and a subsequently unshielded Earth was discussed to explain the isotopic anomalies of [60]Fe on Earth[71–73]. It was assumed that the compression of the heliosphere was either induced by a near-Earth supernova[72,72,73] or by the passage of the solar system through a dense cold cloud[71]. The compression of the heliosphere beyond Earth's orbit would lead to an increased GCR flux by a factor of more than 4[71]. The required distance of the near-Earth supernova was estimated to be within 10 pc, close to or within the so-called kill-radius[74,75], the distance a supernova would lead to cataclysmic changes in Earth's atmosphere and biosphere[76,77]. A compression of the heliosphere by a supernova was therefore excluded for the 2–3 Myr ago period of a discovered supernova-produced [60]Fe influx[72,73] and similarly can be excluded for the time period of the [10]Be anomaly. Previous findings of interstellar [60]Fe due to near-Earth supernovae do not show a corresponding peak in [10]Be. Although it is difficult to detect [60]Fe in the time period of the [10]Be anomaly due to advanced decay, there is no indication for a concomitant spike occurrence of these two radionuclides so far[31]. This is in agreement with the hypothesis that the detected [60]Fe is condensed into interstellar dust, whereas [10]Be is

produced by cosmic rays. The supernova yield of [10]Be is negligible compared to cosmogenic production[31,78].

An increased GCR flux around 2–3 Myr ago due to the solar system's encounter with a cold cloud[71] can be excluded by the non-anomalous [10]Be concentrations in a variety of deep-ocean samples, subject of a subsequent publication. However, the here-reported [10]Be anomaly as a consequence of an increased GCR flux would be in agreement with the proposed scenario. The rebound timescale of the heliosphere to its full extension is long, a few 100 kyr[72], also in agreement with the extended [10]Be anomaly. Future modelling of the solar system's trajectory with respect to potential dense clouds during the time period of the [10]Be anomaly is direly required.

The solar system revolves around the galactic centre but also oscillates perpendicularly to the galactic plane[79]. This z-oscillation was discussed as a reason for mass extinction events and it being imprinted into Earth's geological record[80,81]. The oscillation half-period is on the order of 30–50 Myr and the last passing of the galactic plane happened about 3 Myr ago[79,82–84]. Therefore, the z-oscillation cannot account for the [10]Be anomaly, however, it could be a source for a higher GCR flux[85]. The solar system's revolution and non-zero relative motion with respect to the galactic spiral arms lead to crossings of these high-density regions in the Milky Way. Here, the period for encounter is believed to be on the order of 140 Myr[86,87], again much longer than required to explain the [10]Be anomaly. The impacts of such crossings on Earth's atmosphere and biosphere are, however, disputed[88]. The crossing of higher density regions within the spiral arms, such as the boundary of the Local Bubble[89], happens on more compatible shorter timescales, however, their impact is similarly unclear.

Supernovae or gamma-ray bursts (GRB) are intense sources of cosmic rays in the universe. A near-Earth supernova could enhance the cosmic ray flux in addition to the deposition of interstellar radionuclides. A canonical supernova at a distance of 20 pc with 1% of its kinetic energy going into cosmic rays would double the [10]Be production on Earth[90]. An isotropic near-Earth supernova is unlikely to be the cause of the [10]Be anomaly due to the proximity to the kill radius of 8–20 pc[74,75] and the low density of candidate stars around the solar system. If the emission of the cosmic rays is anisotropic and directed towards Earth or the cosmic ray intensity is higher than expected due to a higher luminosity or more efficient cosmic ray transport, a more distant supernova could still be a viable scenario. A potentially cosmic ray-induced anomaly on Earth a few Myr prior to the arrival of [60]Fe-containing supernova-produced dust, however, is intriguing and requires further investigations to exclude a causal connection. Another high-energy scenario is GRB with their emission cone pointing towards Earth. It was hypothesised that GRB could strongly enhance cosmogenic production of [14]C or [10]Be by several orders of magnitude on Earth[91]. However, regular GRB last on the order of seconds up to minutes for long-duration GRB[92]. Consequently, a [10]Be anomaly over several 100 kyr to Myr is excluded to stem from directed cosmic radiation from GRB.

In summary, the compression of the heliosphere by the solar system's encounter with a cold cloud or a complex supernova are reasonable scenarios to explain an enhanced production of [10]Be on Earth.

## Future detections

The here-reported [10]Be anomaly during the late Miocene was conclusively discovered in several Pacific deep-ocean ferromanganese crusts. The most promising origin of the anomaly is either a grand re-organization of oceanic circulation with the onset and ramp-up of the Antarctic Circular Current as a terrestrial origin, or the temporary enhancement of the galactic cosmic ray flux through a near-Earth supernova or the compression of the heliosphere by the passage through a cold cloud as astrophysical origins. This anomaly was detected in the Central and the Northern Pacific. Due to the long

residence time of $^{10}$Be in the water column, which is on the order of hundreds to one thousand years[15,93,94] and similar to ocean circulation times[95,96], the anomaly should be present throughout the Pacific. Two important open questions need to be addressed in future investigations: Is this $^{10}$Be anomaly a global phenomenon and what is the exact timing and width of this anomaly? Both questions can be addressed by analysing deep-sea sediments with low sedimentation rates on the order of mm/kyr to reduce diagenetic effects. Sediments have excellent time resolution on the Myr timescale due to their about 1000 times higher sedimentation rate compared to ferromanganese crusts. Furthermore, sediments also regularly and continuously accumulate $^{10}$Be in their matrix. Pacific sediments can be used to further investigate the here reported timing and amplitude, whereas a discovery of the anomaly in a sediment from a different location on Earth would make it a global anomaly and rule out several terrestrial scenarios for the anomaly's origin. Considering a broadening of the anomaly in a ferromanganese crust due to diffusion, pore-water and remobilization, a deep-sea sediment could feature a sharper and thus more pronounced anomaly. The time interval of 9–11.5 Myr ago in sediments should, therefore, be sampled and measured for $^{10}$Be with high time resolution.

The detection of this anomaly with a different cosmogenic (radio) nuclide would provide further information about the origin of this anomaly. However, most of the long-lived radionuclides are not suitable. The stable isotope $^{3}$He is a proxy for the amount of micrometeoritic influx in a sample. A globally increased micrometeoritic influx cannot explain the $^{10}$Be anomaly due to the orders of magnitude higher atmospheric $^{10}$Be flux. The radionuclides $^{26}$Al ($t_{1/2} = 0.7$ Myr), $^{36}$Cl ($t_{1/2} = 0.3$ Myr) and $^{41}$Ca ($t_{1/2} = 0.1$ Myr) are too short-lived to be still detectable at 10.1 Myr ago. Longer-lived $^{129}$I ($t_{1/2} = 16$ Myr) suffers from environmental contamination from nuclear activities as well as a fissiogenic in-situ background. The most promising cosmogenic radionuclide besides $^{10}$Be is $^{53}$Mn ($t_{1/2} = 3.7$ Myr), though its measurement by neutron activation analysis or AMS is still challenging. Advanced AMS facilities incorporating a selective laser photodetachment system[97] such as the Helmholtz Accelerator Mass Spectrometer Tracing Environmental Radionuclides (HAMSTER) might be able to measure $^{53}$Mn in deep-ocean samples with high precision and sensitivity in the future.

## Methods
### VA13/2-237KD
The processing and characterization of the main ferromanganese crust sample of this work was already extensively discussed in[31,32,98]. A 3.7 kg piece of the hydrogenetic ferromanganese crust VA13/2-237KD (Fig. 1) from the Central Pacific (9° N, 146° W) covering more than 20 Myr of geological history was scanned with an optical high-resolution 3D scanner. A digital as well as a physical 3D model was generated. The inner structure of the crust was investigated by a micro-CT scan with 150 μm resolution. Two drill-holes (a) and (b) with 1–2 mm depth resolution were taken resulting in depth profiles with a time resolution of <400 kyr with 400–800 mg per individual sample. The individual samples were chemically processed to extract atmospheric $^{10}$Be. A 30–200 mg aliquot of finely milled crust powder was dissolved in 10.2 M HCl. About 500 μg of stable terrestrial $^{9}$Be with well-characterised low $^{10}$Be content was added to the samples as carrier and chemical yield tracer. Refractory minerals were separated and Be was precipitated as hydroxide together with Fe by the addition of ammonia solution. The precipitate was redissolved in 10.2 M HCl and loaded onto an anion-exchange column, where Be was separated from Fe. The purified Be fraction was transformed to BeO and mixed with high-purity Nb powder 1:4 wt:wt. The naturally abundant $^{9}$Be in the ferromanganese crust is negligible compared to the added $^{9}$Be and was determined by inductively-coupled plasma mass spectrometry (ICP-MS) at the Helmholtz-Zentrum Dresden-Rossendorf (HZDR), Germany and the ALS Water Resources Group in Fyshwick, Australia prior to the addition of the carrier. The resulting highly purified BeO fractions were

loaded into individual Cu cathodes for the AMS measurement of $^{10}$Be. The AMS measurements were carried out at the DREsden AMS (DREAMS) facility of HZDR, a versatile 6 MV tandem accelerator[99,100], particularly suitable for high-precision $^{10}$Be measurements with a high total efficiency[101]. The reference material SMD-Be-12[99] was used to normalize the measured isotopic ratios. The $^{10}$Be concentrations in the ferromanganese crust can be calculated by the measured $^{10}$Be/$^{9}$Be ratio from AMS and the well-known added amount of $^{9}$Be. The background level of processed samples was determined by processing blanks and reflects the intrinsic purity of the added carrier as well as the laboratory and instrument background. The background levels of $^{10}$Be/$^{9}$Be $= 4 \times 10^{-15}$ for a commercial $^{9}$Be solution from ACR[32] and $^{10}$Be/$^{9}$Be $< 5 \times 10^{-16}$ for an in-house prepared $^{9}$Be solution from shielded phenakite mineral[102] are several orders of magnitude lower than the measured $^{10}$Be/$^{9}$Be ratios in the ferromanganese crusts and, therefore, negligible. Any concentration uncertainties are calculated as 1-$\sigma$ confidence levels while horizontal bars indicate the depth or age interval of each sample.

A different, more centrally located piece of the crust VA13/2-237KD was initially dated using cosmogenic $^{10}$Be by Segl et al. in 1984 after retrieval[20]. The profile of Segl et al. has a lower depth resolution of 2 mm until ≤ 34 mm with the last data point being at 34–38 mm. The $^{10}$Be concentrations were determined by AMS at the Zurich tandem accelerator lab[103]. The comparison of absolute $^{10}$Be concentrations requires the normalization to reference material. The absolute values from Segl et al.[20] are likely systematically shifted due to an update in the $^{10}$Be half-life in 2010 and the availability of more accurately and consistently determined reference materials these days. However, a constant surface $^{10}$Be concentration can be used to normalize all profiles of the same crust despite their offsets. Recently, a mathematical lapse was found and corrected in their published data[21].

### SO142-4DR
An incomplete drill-hole profile of the crust SO142-4DR from the Northern Pacific (32° N, 159° W) became available from the now-closed accelerator lab in Munich[104]. This crust was previously used to investigate a potential interstellar $^{53}$Mn influx on Earth[35]. The available material was similarly processed, however, a natural $^{9}$Be determination was not possible due to the limited available sample mass and low natural $^{9}$Be concentrations. The determined $^{10}$Be concentration profile can, however, be compared to the more precise profiles of VA13/2-237KD.

### Crust-3
Wallner et al.[33] reported on the detection of interstellar radionuclides in a ferromanganese crust. They similarly dated their ferromanganese crust (Crust-3, Pacific, exact location unknown due to resource protection, 17° N, 170° W to 19° N, 167° E) by atmospheric $^{10}$Be at the MALT AMS facility[105], however, only with six individual highly depth-resolved samples. Any features of this profile can, therefore, only be seen as indicative to support more precise data.

## Data availability
The $^{10}$Be data generated in this study are provided in the Supplementary information file.

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

## Acknowledgements

D.K. was supported by an AINSE Ltd. Postgraduate Research Award (PGRA). This work was supported by the Australian Research Council's Discovery scheme, project numbers DP180100495 and DP180100496 (A.W.) and through RADIATE (824096) from the EU Research and Innovation program HORIZON 2020, project numbers 21002421-ST and 20002142-ST (D.K.). This research was carried out at the Ion Beam Center (IBC) at the Helmholtz-Zentrum Dresden-Rossendorf e.V., a member of the Helmholtz Association, project number 22003072-EF (D.K.). The authors want to thank Gunther Korschinek for generously providing the crust SO142-4DR and his help to organise the large piece of VA13/2-237KD.

## Author contributions

This work is a result of the PhD Thesis of D.K.[31]. D.K., J.L. and A.W. wrote the manuscript, and all authors were involved in the project and commented on the paper. D.K. initiated and designed the study. A.W. acquired the crust sample VA13/2-237KD. D.K. acquired the crust sample SO142-4DR. D.K., S.F., S.M. and Z.S. prepared the samples for AMS and ICP-MS. S.B. performed the ICP-MS measurements at HZDR. D.K., J.L. and G.R. performed the AMS measurements at HZDR with support from S.F., S.M., C.V.V., S.W. and A.W. D.K. performed the data analysis. D.K., J.L. and A.W. interpreted the data.

## Funding

## Competing interests

The authors declare no competing interests.
