## [Transparent Peer Review file · Nature Communications]

A cosmogenic ^{10}Be anomaly during the late Miocene as independent time marker for marine archives

Corresponding Author: Dr Dominik Koll

Version 0:

Reviewer comments:

Reviewer #1

(Remarks to the Author)

Review of NCOMMS-24-63568 "A cosmogenic ^{10}Be anomaly during the late Miocene as independent time marker for marine archives" by Brian C. Thomas

This article presents measurements of a ^{10}Be anomaly in deep ocean ferromanganese crusts from multiple Pacific locations.

This is an important contribution to the field.

The article is well written and presents sufficient data and analysis to establish the discovery of this ^{10}Be anomaly. The authors do an excellent job of presenting possible explanations and evaluating the likelihood of those explanations. The discovery in multiple locations indicates this is not a local effect, but could be regional (rather than global). The authors acknowledge that they have not established that this is a global signal and note that resolving that issue would shed light on possible explanations (ie. terrestrial vs cosmic). I agree with their assessment that a heliospheric compression event is a good candidate to explain the anomaly. A supernova explanation is harder to reconcile, due to the lack of a ^{60}Fe signal at this time. However, it may also be expected that a ^{60}Fe signal would be very difficult to detect here given the age, half-life of ^{60}Fe , and expected deposition amount.

My only suggestion for revision would be for the authors to consider adding more discussion (briefly) of the supernova ^{60}Fe issue noted above.

Otherwise, I recommend publication at this time.

Reviewer #2

(Remarks to the Author)

This short article reports on a prolonged a cosmogenic ^{10}Be anomaly during the late Miocene recorded in several Pacific deep-ocean ferromanganese crusts in the time period 9 – 11.5 Myr ago peaking at 10.1 Myr. Potential origins of this anomaly are discussed.

The results are robust and well presented : text and figures are clear and well documented (more than 100 refs are provided to support the techniques, methods and discussion.

I recommend the publication after minor revisions listed herebelow.

Discussion lines approx. 37 -38 « Paleomagnetic records indicate a slightly lower geomagnetic field but show no extraordinary clustering of geomagnetic events during the period of the ^{10}Be anomaly in the late Miocene. »

Too short discussion !! " no extraordinary" does not mean anything.

Repeated geomagnetic dipole lows linked to reversals and/or crypto-events (such as geomagnetic excursions of the same type as Laschamp) may well be able to trigger your observation.

You cannot dismiss such an hypothesis and you should on the contrary propose to open such a perspective of research for paleomagnetists working on paleointensities of lava flows , on magnetic anomalies of the ocean floor...

The conclusion is too much an abstract.

Note that in sediments sequences, beyond certain depths, potential diagenetic problems can affect (erase) the Be-10 signatures... the search for the 10.1 event should thus be made in relatively low sed rates (few mm/kyr).

Details :

- Discussion lines approx. 25-26:

The peak of the anomaly in drill-hole (a) is fitted by a Gaussian to 10.1 Myr with a FWHM precise : Full Width at half maximum (FWHM)

- Figure 2 and 4 provide the same black and red marks... you may replace fig. 2 by fig. 4.

At least it would be logical to put fig. 4 just after fig. 2 and move figure 3 (on the time scale) as fig. 4.

- Figure 5 : why is the right foot of the blue curve not passing through the data points?

Reviewer #3

(Remarks to the Author)

General comments:

The manuscript 'A cosmogenic ^{10}Be anomaly during the late Miocene as independent time marker for marine archives' by Koll et al. presents a study of ^{10}Be concentrations in ferromanganese crusts from the North Pacific. Final aim is the establishment of a new chronological tie-point, marked by enhanced ^{10}Be concentrations 9 to 11.5 Myr BP.

The efforts undertaken are methodologically state of the art and the results provide robust indications for unusually high ^{10}Be concentrations in North Pacific ferromanganese crusts during that period. Therefore, the study is of interest for a broader geoscience community and suitable for publication in Nature Communications. However, before publication two important points should be developed and discussed in more detail. The lack of line numbers complicates the writing of the review.

Important points:

Two points that should be developed are (i) the spatial relevance of the ^{10}Be excursion and (ii) the discussion about its underlying mechanism (ocean circulation changes, compression of the heliosphere/complex supernova). In general, I think that the discussion about the origin of the ^{10}Be excursion should focus in more detail on the likely reasons, while the text about the unlikely causes (Miyake Events, solar activity, geomagnetic field strength) can be shortened.

Spatial relevance (ocean circulation):

The presented records of the ^{10}Be anomaly are from ferromanganese crusts in the North Pacific only, and changes in ocean circulation paralleled by the onset of the modern Antarctic Circumpolar Current are discussed as one of the likely causes for the observed increase in ^{10}Be 9.5-11 Myr ago. Because of the coinciding well-documented major reorganizations in ocean circulation this alternative seems most plausible for me. However, combined with the confined sample area, changes in ocean circulation would only allow to postulate a ^{10}Be anomaly in the North Pacific. This would restrict the use of the ^{10}Be anomaly as chronological tie-point and should be clarified in the text and title of the manuscript.

Compression of the heliosphere/complex supernova:

Further likely mechanism for the ^{10}Be anomaly is increased GCR flux due to an encounter with a cold cloud or complex supernova. This would result in a global increase in ^{10}Be production and requires further investigation. Although existing data from other ferromanganese crust were mentioned in the manuscript (nearly all do not cover 9-11.5 Myr BP), a literature review about possible cosmogenic radionuclide signals in marine sediment cores from other ocean basins is missing. Maybe this could help to investigate both hypotheses.

Specific comments:

(1) Text (introduction): 'The residence time of ^{10}Be in the atmosphere is on the order of days to weeks until it attaches to aerosols and precipitates'.

To my knowledge ^{10}Be has an atmospheric residence time of 1 to 2 years (e.g. Raisbeck et al., 1981).

(2) Ref 19 (introduction):

Ref 19 proposes possible future applications. Was ^{10}Be decay-dating really applied in ice cores?

(3) Text (introduction): Well-known deviations from constancy such as the Laschamps event [24, 25] or Miyake-events [26,

27].

Please add the durations: Laschamp (multi-centennial); Miyake (one year).

(4) Text (introduction): Several deep-ocean samples showed so far fairly constant ^{10}Be levels through their ^{10}Be concentration as well as $^{10}\text{Be}/^9\text{Be}$ ratio profiles over the last 10 Myr [14, 15, 20–23, 30].

In an uncommented way this introduction sentence feels a bit confusing since 10 Myr lies right in the middle of the discussed ^{10}Be excursion (9–11.5 Myr BP). Additionally, this could point to a spatial restriction of the ^{10}Be anomaly to the North Pacific.

(5) Please add a map to the manuscript showing the sampling locations to the manuscript, maybe as Fig. 1? This would allow to more easily follow the text.

(6) Methods (text): Two drill-holes (a) and (b) with 1 - 2 mm depth resolution were taken resulting in high resolution depth profiles.

Please add the total amount of samples per drill-hole and the approx. time covered by one sample.

(7) Here, we report on the discovery of an anomaly in the ^{10}Be concentration profiles of several deep-ocean ferromanganese crusts during the late Miocene.

Add 'North Pacific' to the sentence.

Detailed comments:

Page 5: Replace 'time-correlated' with 'synchronous'.

Provide the long version of FWHM at first use.

made.

Reviewer#1:

The article is well written and presents sufficient data and analysis to establish the discovery of this ^{10}Be anomaly. The authors do an excellent job of presenting possible explanations and evaluating the likelihood of those explanations.

Thank you very much for this positive feedback!

The discovery in multiple locations indicates this is not a local effect, but could be regional (rather than global). The authors acknowledge that they have not established that this is a global signal and note that resolving that issue would shed light on possible explanations (ie. terrestrial vs cosmic).

Thank you for pointing out that we do not claim a global signal but rather discuss the possibility and the consequences of a global signal with respect to the potential origin of the anomaly.

I agree with their assessment that a heliospheric compression event is a good candidate to explain the anomaly. A supernova explanation is harder to reconcile, due to the lack of a ^{60}Fe signal at this time. However, it may also be expected that a ^{60}Fe signal would be very difficult to detect here given the age, half-life of ^{60}Fe , and expected deposition amount.

We indicated that it would be intriguing if the ^{10}Be anomaly was linked to the ^{60}Fe discoveries in deep-ocean archives. Such a link would require sophisticated astrophysical modelling of cosmic ray propagation from a near-Earth supernova in addition to a potentially delayed ^{60}Fe -containing-dust transport. We look forward to future research investigating this possibility.

My only suggestion for revision would be for the authors to consider adding more discussion (briefly) of the supernova ^{60}Fe issue noted above.

Done, a brief discussion was included.

Otherwise, I recommend publication at this time.

Thank you very much for your review.

Reviewer#2:

This short article reports on a prolonged cosmogenic ^{10}Be anomaly during the late Miocene recorded in several Pacific deep-ocean ferromanganese crusts in the time period 9 – 11.5 Myr ago peaking at 10.1 Myr. Potential origins of this anomaly are discussed. The results are robust and well presented : text and figures are clear and well documented (more than 100 refs are provided to support the techniques, methods and discussion).

I recommend the publication after minor revisions listed here below.

Thank you very much for your review and your valuable comments.

Discussion lines approx. 37 -38 « Paleomagnetic records indicate a slightly lower geomagnetic field but show no extraordinary clustering of geomagnetic events during the period of the ^{10}Be anomaly in the late Miocene. »

Too short discussion !! "no extraordinary" does not mean anything.

Repeated geomagnetic dipole lows linked to reversals and/or crypto-events (such as geomagnetic excursions of the same type as Laschamp) may well be able to trigger your observation.

You cannot dismiss such an hypothesis and you should on the contrary propose to open such a perspective of research for paleomagnetists working on paleointensities of lava flows , on magnetic anomalies of the ocean floor...

We agree with the reviewer's assessment that a search for magnetic anomalies should be proposed. We weakened our statement from excluding geomagnetism as a cause for the anomaly to consider it being unlikely but requiring further investigations.

The conclusion is too much an abstract.

The editorial suggestion is to remove the conclusion section and integrate it into the results section. We re-wrote the paragraph keeping the suggestion of the reviewer in mind.

Note that in sediments sequences, beyond certain depths, potential diagenetic problems can affect (erase) the Be-10 signatures... the search for the 10.1 event should thus be made in relatively low sed rates (few mm/kyr).

Thank you very much for this note. We added to the sediment proposal "with low sedimentation rates on the order of mm/kyr to reduce diagenetic effects"

Details :

- Discussion lines approx. 25-26:

The peak of the anomaly in drill-hole (a) is fitted by a Gaussian to 10.1 Myr with a FWHM precise : Full Width at half maximum (FWHM)

Done.

- Figure 2 and 4 provide the same black and red marks... you may replace fig. 2 by fig. 4.

At least it would be logical to put fig. 4 just after fig. 2 and move figure 3 (on the time scale) as fig. 4.

We moved Fig 2 into the supplementary to avoid showing the data twice.

- Figure 5 : why is the right foot of the blue curve not passing through the data points?

The single Gaussian fit correctly considers the oscillation from 0-9 Myr as the regular baseline fluctuation in contrast to the enhancement of the anomaly. The oscillation from 0-9 Myr could point to an additional feature. The right foot of the fit, therefore, still considers the data mostly as top part of the baseline consistent with the interval 0-9 Myr such that the fit converges. The decay-correction of data older than the anomaly is difficult due to a gradual flattening of the profile, most likely caused by a structural change of the ferromanganese crust.

Reviewer#3:

General comments:

The manuscript 'A cosmogenic ^{10}Be anomaly during the late Miocene as independent time marker for marine archives' by Koll et al. presents a study of ^{10}Be concentrations in ferromanganese crusts from the North Pacific. Final aim is the establishment of a new chronological tie-point, marked by enhanced ^{10}Be concentrations 9 to 11.5 Myr BP.

The efforts undertaken are methodologically state of the art and the results provide robust indications for unusually high ^{10}Be concentrations in North Pacific ferromanganese crusts during that period. Therefore, the study is of interest for a broader geoscience community and suitable for publication in Nature Communications. However, before publication two important points should be developed and discussed in more detail. The lack of line numbers complicates the writing of the review.

Thank you very much for your review and you detailed comments that improve the manuscript.

Important points:

Two points that should be developed are (i) the spatial relevance of the ^{10}Be excursion and (ii) the discussion about its underlying mechanism (ocean circulation changes, compression of the heliosphere/complex supernova).

See further below.

In general, I think that the discussion about the origin of the ^{10}Be excursion should focus in more detail on the likely reasons, while the text about the unlikely causes (Miyake Events, solar activity, geomagnetic field strength) can be shortened.

We think that the text about unlikely causes is relatively concise considering the many options discussed.

Spatial relevance (ocean circulation):

The presented records of the ^{10}Be anomaly are from ferromanganese crusts in the North Pacific only, and changes in ocean circulation paralleled by the onset of the modern Antarctic Circumpolar Current are discussed as one of the likely causes for the observed increase in ^{10}Be 9.5-11 Myr ago. Because of the coinciding well-documented major reorganizations in ocean circulation this alternative seems most plausible for me. However, combined with the confined sample area, changes in ocean circulation would only allow to postulate a ^{10}Be anomaly in the North Pacific. This would restrict the use of the ^{10}Be anomaly as chronological tie-point and should be clarified in the text and title of the manuscript.

We agree with the reviewer that the reorganization of the ocean circulation might be a well-fitting explanation for the anomaly. We added a short paragraph to further emphasize this point.

However, we disagree that the samples in our work restrict our statement to the North Pacific. The two ferromanganese crusts were located 9°N, 146°W (Central Pacific) and 32°N, 159°W (Northern Pacific). The distance between these two samples is almost 3000 km. The residence time of ^{10}Be in the ocean is several hundred years up to 1000 years. The ocean ventilation time of the Antarctic Bottom Water to the northern limit of the Pacific Ocean is about 1000 years [Matthew England, JPO 25 (1995)]. In comparison, the ventilation time to the Southern Ocean is about 100 years. The global deep-water ocean circulation timescale is about 1000 years [Jönsson and Watson, Nat Comm. 7:11239 (2016)]. Given the similar timescales of ^{10}Be residence time and ocean ventilation and circulation times as well as the detection of the anomaly in the Central Pacific as well as the Northern Pacific, at least a Pacific-wide anomaly needs to be considered. By restricting the papers statement to the North Pacific, we might even purport a certainty about a regional cause or effect, while the origin of the ^{10}Be enhancement could lie in the southern hemisphere. In light of the given arguments and the clearly stated mere “potential” for a global anomaly (as acknowledged by reviewer #1), we keep the title of the paper. The abstract is rewritten now stating that the ferromanganese crusts are from the Central and the Northern Pacific. A paragraph in the newly written conclusion of the Discussion section about a regional, Pacific-wide or global anomaly and how to discern these possibilities with sediments or other radionuclides is added.

Compression of the heliosphere/complex supernova:

Further likely mechanism for the ^{10}Be anomaly is increased GCR flux due to an encounter with a cold cloud or complex supernova. This would result in a global increase in ^{10}Be production and requires further investigation. Although existing data from other ferromanganese crust were mentioned in the manuscript (nearly all do not cover 9-11.5 Myr BP), a literature review about possible cosmogenic radionuclide signals in marine sediment cores from other ocean basins is missing. Maybe this could help to investigate both hypotheses.

We very much agree that other cosmogenic signals would be very important to distinguish between a higher production rate of ^{10}Be or a terrestrial reason for the anomaly. Unfortunately at the timescale of the anomaly (about 10 Myr), the options for other radionuclides are very limited. ^{26}Al (717kyr), ^{36}Cl (301kyr) and ^{41}Ca (99kyr) are too short-lived to be still detectable; lowest levels of longer-lived ^{129}I (16Myr) face serious environmental contamination problems from nuclear weapons tests, reprocessing plants and natural fissiogenic ^{129}I . To our knowledge, the last remaining viable radionuclide candidate is ^{53}Mn (3.7Myr), where the database is very limited due to the difficulty of measuring low-levels of ^{53}Mn with neutron-activation analysis or AMS. To our knowledge, the literature does not offer any further data. We added a statement in the newly written conclusion of the Discussion section. We want to note that ^3He is cosmogenic but stable and a proxy for micrometeorites. The option of an increased micro meteoritic flux was not discussed in the work because of the unreasonably high enhancement required to outweigh an atmospheric ^{10}Be signal. A brief discussion about this is also added to the manuscript.

Specific comments:

(1) Text (introduction): 'The residence time of ^{10}Be in the atmosphere is on the order of days to weeks until it attaches to aerosols and precipitates'.

To my knowledge ^{10}Be has an atmospheric residence time of 1 to 2 years (e.g. Raisbeck et al., 1981).

Very well spotted. Thank you for that. The residence time of ^{10}Be in the atmosphere is indeed 1-2 years. The here mentioned weeks are the tropospheric residence time. Corrected.

(2) Ref 19 (introduction):

Ref 19 proposes possible future applications. Was ^{10}Be decay-dating really applied in ice cores?

We state that ^{10}Be dating is regularly applied to ice cores (ref 19). This can be decay dating for old ice, but more regularly uses excursions and patterns together with radionuclides such as ^{14}C or ^{26}Al .

See Horiuchi et al. Quaternary Geochronology 3 (2008) as an example.

(3) *Text (introduction): Well-known deviations from constancy such as the Laschamps event [24, 25] or Miyake-events [26, 27].*

Please add the durations: Laschamp (multi-centennial); Miyake (one year).

Done.

(4) *Text (introduction): Several deep-ocean samples showed so far fairly constant ^{10}Be levels through their ^{10}Be concentration as well as $^{10}\text{Be}/^9\text{Be}$ ratio profiles over the last 10 Myr [14, 15, 20–23, 30].*

In an uncommented way this introduction sentence feels a bit confusing since 10 Myr lies right in the middle of the discussed ^{10}Be excursion (9-11.5 Myr BP). Additionally, this could point to a spatial restriction of the ^{10}Be anomaly to the North Pacific.

Agreed. We modified the sentence to several Myr to avoid any confusion.

(5) *Please add a map to the manuscript showing the sampling locations to the manuscript, maybe as Fig. 1? This would allow to more easily follow the text.*

Very much agree, thank you. Added.

(6) *Methods (text): Two drill-holes (a) and (b) with 1 - 2 mm depth resolution were taken resulting in high resolution depth profiles.*

Please add the total amount of samples per drill-hole and the approx. time covered by one sample.

Done.

(7) *Here, we report on the discovery of an anomaly in the ^{10}Be concentration profiles of several deep-ocean ferromanganese crusts during the late Miocene.*

Add 'North Pacific' to the sentence.

We modified the sentence to: Here, we report on the discovery of an anomaly in the ^{10}Be concentration profiles of several deep-ocean ferromanganese crusts from the Central and Northern Pacific during the late Miocene.

Detailed comments:

Page 5: Replace 'time-correlated' with 'synchronous'.

Provide the long version of FWHM at first use.

Both done.